# Analysis of Touristification Processes in Historic Town Centers: The City of Seville

Germán Herruzo-Domínguez *, José-Manuel Aladro-Prieto and Julia Rey-Pérez

University Institute of Architecture and Building Sciences, School of Architecture, University of Seville, Av. Reina Mercedes 2, 41012 Seville, Spain; aladroprieto@us.es (J.-M.A.-P.); jrey1@us.es (J.R.-P.)
* Correspondence: gherruzo@us.es

**Abstract:** Encouraged by the administration, the gen9otrification process has been useful in economic terms for the reactivation of the socio-cultural fabrics of historic urban spaces in decline. What was initially considered an advantage has led to the touristification of historic centers, and in turn to the alteration of their original use. In these settings, the demographic void caused by increasingly shunning local identity has combined with pressure from excessive tourism and the obsolescence of heritage protection bodies in charge of conservation. Given the crisis affecting the definitions of the current system, this study aims to review the environmental agents of heritage value in relation to the processes of touristification and gentrification. Data obtained from different methodologies are analyzed using a multidisciplinary database, a model which enables the analysis of the relevant information from the different interacting fields. This case study focuses on the historic town center of Seville, specifically between 2015 and 2020. Elements are defined as indicators for these processes and the analysis of this case study will comprise the main results of this research.

**Keywords:** gentrification; touristification; tourism in historic cities; heritage protection; Seville



## 1. Introduction

The current socio-cultural panorama focuses on the promotion of policies—both national and local—which encourage the promotion of tourism as one of the main economic driving forces in heritage cities [1]. However, this is accompanied by major concern about the impact of cultural tourism, with a growing number of consequences as time progresses, as can be seen from the actions of bodies such as the European Parliament [2]. The influence of the rise of mass tourism has not only been reflected in major European capitals such as Barcelona, Athens, Lisbon, Rome, or Berlin [3–5], but this phenomenon has also reached America [6–8]. Nevertheless, tourism not only focuses on these major cities, but tourist flows are beginning to intensify in economically disadvantaged areas as a potential economic focal point. The infrastructural renewal of tourism also becomes particularly interesting, through online tourist accommodation, low-cost airlines, and cruise tourism [9]. We are facing a global phenomenon [10] that is beginning to be re-evaluated in terms of both theoretical and practical issues [11], and efforts are being made to address it through sustainable proposals [12] or "green gentrification" [13].

At the outset, an element we find closely linked to these processes is gentrification, the process of urban transformation in neighborhoods—usually historic ones. This situation, which tends to be linked to socio-economic conditioning factors, specifically related to the immediate contexts, can be studied in general terms [14]. Thus, gentrification could be defined as a process of revalorization of an urban sector within the city, usually the historic center, prompting a movement for demographic renewal in which a neighborhood in a state of decline comes to house residents with a higher spending power, or more competitive uses, usually linked to ecommerce [15]. This also leads to issues in terms of identity [16], as the working classes and traditional activities eventually disappear or—at best—move to expansion zones in the city's periphery, as seen throughout the 20th century [17].

In addition, the concept of 'touristification' arises. This concept, which has gained widespread prominence in recent years, emphasizes a gentrification that is far more localized in terms of its intent. According to Jover and Díaz Parra [18], we speak of touristification when describing gentrification in city centers where the origin and end of the process is tourism in and of itself, while tourists are considered the social entities which replace the population previously occupying this urban fabric. A major distinction can be made between both concepts as gentrification does not necessarily imply a depopulation process caused by occasional settlement, but touristification risks focusing land use on consumers of temporary leisure, rootless and in constant movement [19]. It also interferes in the land market, increasing speculation within the local market and creating a rental market which is geared towards the sometimes-excessive revalorization of certain parts of the city [19]. This ultimately brings about a situation of inequality, wearing down residents who have traditionally occupied these areas and who eventually admit defeat when faced with financial interests seeking to dedicate built-up land to almost a single use: tourism [20,21].

It is at this point that the role of short-term rentals comes into place, with platforms such as Airbnb and Homeaway increasing the number of these rentals in recent years [22]. The high demand for tourist accommodation in recent decades, which hotel facilities in the city are no longer able to accommodate, has led to these becoming obsolete [23].

The particular case of Seville provides us with an interesting case study [20,24,25] as it is a heritage city with a robust economic policy based on tourism, which is one of its financial pillars. With a population of 681,988 for the year 2022 according to the most recent census for the Spanish National Statistics Institute, there is an average monthly flow of 170,000 temporary visitors [26]. This amounts to almost one-third of the population residing full time in the city. The city must therefore be equipped to offer tourist accommodation to a high number of people in a very limited timespan, which hotels are ill equipped to cope with. In view of the above, the short-term rental infrastructure has increased its prominence in the city, where there were over 10,000 tourist apartments in the year 2018 [27].

The final result of all the above, specifically in terms of the Ayuntamiento de Sevilla [Seville Town Council] and above all the Gerencia de Urbanismo [Town Planning Department], is the existence of an obsolete General Plan which remains practically unchanged since it was passed in 2006, except for the odd modification. Due to a lack of planning, it fails to provide sufficient protection to the current urban fabric, where an intense process of tourist gentrification and depopulation of the city center is taking place. The value of sustainability, understood as the combination of cultural, social, economic and environmental agents, is in jeopardy and possibly even at risk of disappearing in the face of a situation of decline.

In short, these agents are linked to the interactions and relations between individuals and groups in a given society (social); those linked to the economy and factors which affect the economic prosperity and wellbeing of a population (economic); and those referring to defining elements of the culture and identity of a community (socio-cultural).

This article aims to assess the potential risks of touristification in the city center, focusing specifically on the concept of sustainability in economic, social and cultural terms in a short time span in the recent past. A series of final conclusions will be extracted from an analysis carried out on these three blocks to evaluate the relationship between the evolution of the agents reviewed and a possible gentrification process.

## 2. Methodology

An individual analysis was carried out on three of the four aspects which currently make up the value of heritage sustainability in the district of Casco Antiguo. This value rests on four aspects in total: social, economic, cultural and environmental. This study will focus on the first three aspects, examining a 5-year period in the recent past, from 2015 to 2020, using any available and up-to-date data from different sources for analysis. It will focus on the district of the Casco Antiguo, one of the city's most representative

enclaves, where greater social movement has been detected, making the analysis of the current dynamics and relevance to the current context in the city of particular interest.

Open source ArcGIS 10.8 software is used to enter the different data combined in a model created for case study analysis [28]. The data obtained for the analysis of social aspects was mostly from the Instituto Nacional de Estadística—INE [Spanish Statistical Office] [24] whose data coincide with those found in the Instituto de Estadística y Cartografía de Andalucía—IECA [Institute of Statistics and Cartography of Andalusia], as well as the Municipal Census and Municipal Register for the city of Seville, made available by the Statistics Service of the Ayuntamiento de Sevilla [29]. Finally, the demographic data for the different neighborhoods in the city were obtained from the Statistical Analysis of the city of Seville, also carried out by the Ayuntamiento de Sevilla [30]. This last source is particularly useful as it breaks down demographic data by district and neighborhood, enabling the superimposition of any data analyzed and the real situation as observed in this case study, while the other sources were used for the comparison with the demographic total of the city. However, the timescale is problematic as it covers the period between 2013 and 2017, the year of its publication. These data were used for the analysis of demographic movement within the neighborhood, applying a hypothesis to the analysis of the development of processes in which factors including touristification act upon it, causing these demographic movements.

Data for economic analysis were obtained from the Statistical Yearbook for the city of Seville, by the Ayuntamiento de Sevilla from 2016 to 2020 [29]. In this case, data relating to the average income per person and per neighborhood were obtained for the period between 2015 and 2018. Furthermore, the values for land sale and rental compared to other surrounding areas are also relevant as quantifiable data for the speculation which has appeared in recent years. The information accessed covers the period between 2016 and 2020.

The socio-cultural aspect, viewed here from the angle of appropriation and identity of residents in relation to the neighborhood was studied taking into consideration the social factor as both are closely linked. The offer for rental accommodation was analyzed and compared to residences occupied full time by the population, quantifying the current weight of occupational demand of short-term accommodation and its effect on both the area and permanent residents. Data were extracted from open source portal Datahippo [27] which offers information up to 2018 for all the tourist rental advertisements on offer on the main search platforms (Airbnb, Homeaway). This was complemented with data from the Instituto Nacional de Estadística [26], including the number of tourists visiting the city. Furthermore, information relating to identity and appropriation of the area was provided by numerous associations and collectives collaborating in citizen participation activities. Data at the district level were considered to represent all neighborhoods within it.

The main Assets of Cultural Interest, as well as the three World Heritage Assets found in the city (Alcázar, Cathedral and Archivo de Indias), were studied at the district level. These are considered to be the main foci attracting tourism.

Three individual analyses were interconnected to extract general results as it is recognized that some parts of the process intervene in others, indirectly or directly.

## 3. Results

### 3.1. Demographic Analysis

The demographic analysis shows a tendency in the city of Seville of a decreasing resident population. Between 2015 and 2020, total population decreased by only 0.36% (693,878 inhabitants in 2015 to 691,395 in 2020). However, the year 2019 shows a minimum peak, albeit with a downward trend, with 688,592 inhabitants, while the following year saw a slight demographic increase [26]. On a smaller district scale, no major movement was observed in 2016 and 2017, the only periods which could be obtained through the document for Statistical Analysis for the city of Seville [27]. However, this was not the case for the districts of Sevilla Este—Alcosa—Torreblanca and Macarena, all of which gained approximately 1000 inhabitants in the space of a year.

Nevertheless, specific analysis was carried out at neighborhood level within the Casco Antiguo district in order to see the population movements for 2016 and 2017. Although few modifications were observed notable changes were apparent in intervals for every 40–50 inhabitants lost. This did not apply to neighborhoods such as Museo (20), Alfalfa (90) and Encarnación—Regina (40) where the resident population increased in these years. It should be noted however that the above information does not provide relevant information on gentrification so that the migratory pressure of these neighborhoods, also included in the Statistical Analysis for the city of Seville, was used as reference instead. This value shows the percentage of foreign visitors in relation to the total resident population (Figure 1). In the case of the district of Casco Antiguo, this value is 9%, four points higher than the average for Seville. Despite not falling within the period studied, as records were available from the year 2012 onwards, it was possible to forecast the trend in later years.

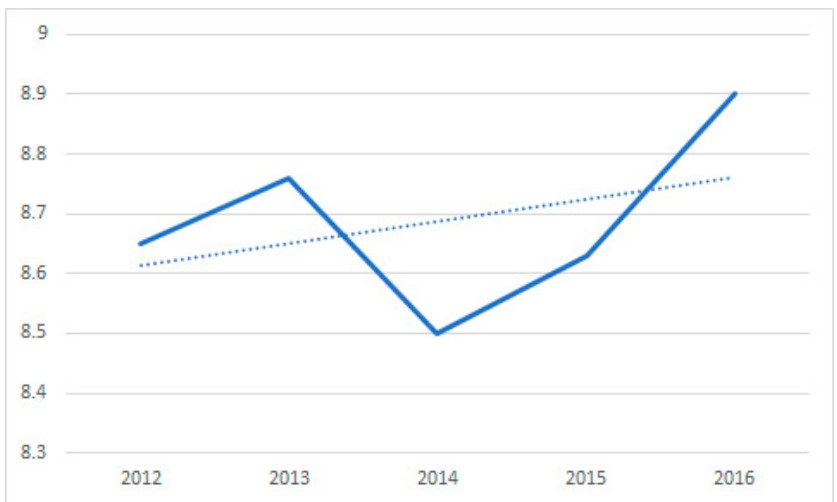

**Figure 1.** Percentages of migratory pressure on the Casco Antiguo district. Source: the author, based on data obtained from [30].

Figure 1 shows an upward trend line, showing that the district will be liable to an increase in pressure from the migrant population establishing their residence in these city center neighborhoods in years to come.

*3.2. Economic Analysis*

An initial economic assessment was carried out on the average per-capita income in the different neighborhoods of the Casco Antiguo district from 2015 to 2018 (Figure 2). A series of significant trends and variations were identified based on the evolution of the values compiled by the Statistics Service [29,30]. Relatively homogeneous income values were observed for the year 2015, with an average of 15,000 euros per person. However, this was clearly an upward trend, with values increasing in all the neighborhoods. Substantial growth was recorded for 2017, with values ranging between 16,065 and 20,177 euros per person. For the final period analyzed, 2018, a final upward trend could be observed, with increases in all except in the neighborhood of San Bartolomé, where it occasionally decreased, and also in the neighborhood of Museo, where a peak disrupted the trend for the other years and neighborhoods. These data point to a gradual increase in the wealth of the central neighborhoods of this district, despite lingering differences between both of these neighborhoods.

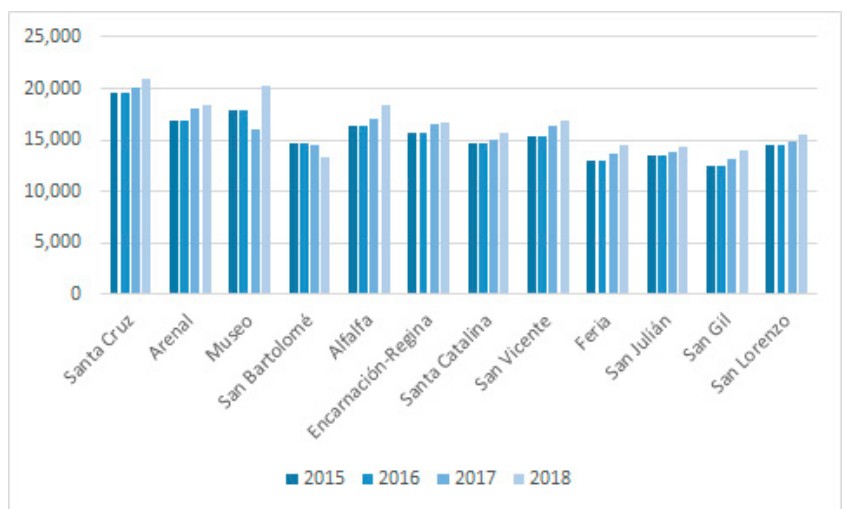

**Figure 2.** Per-capita income in the Casco Antiguo district. Source: the author, based on data obtained from [29].

Furthermore, land sale value was also revised using the data available from the Statistics Service, originally obtained from the Fotocasa website. In order to calculate the data for analysis the annual average for land value was calculated, based on monthly data obtained and calculated as EUR/m$^2$ for the general value of the district of Casco Antiguo. The graph below shows the evolution and trend of the sales value from 2016 to 2020 (Figure 3).

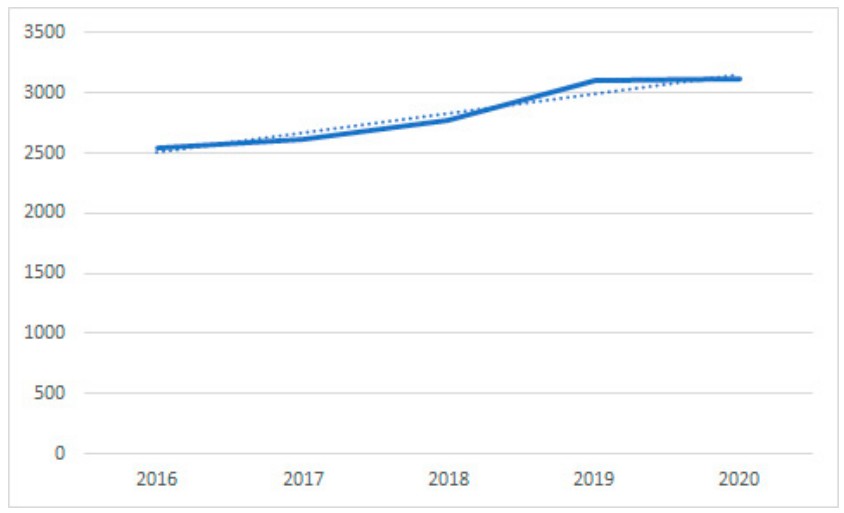

**Figure 3.** Value of land sale in the Casco Antiguo district. Source: the author, based on data obtained from [29].

The price per square meter was 2546.28 EUR in the year 2016, with progressive increases in value recorded until 2020, when it reached the amount of 3118 EUR. Overall, compared with the rest of the city, the average land sale value for Seville was 2091 EUR/m$^2$ in 2020. This shows how the value in the city's historic center exceeded the average value by one-third, which is represented as an example of inflation in the property market.

In terms of the rental market, the average in Seville for the same period ranged between 7.45 EUR/m$^2$ in 2016 and 9.73 EUR/m$^2$ in 2020 (Figure 4). However, the Casco Antiguo district again showed higher values compared with the rest of the city. Figure 4 shows how the rental values in this district, also recorded as an annual average in the Statistics Service and obtained from the Fotocasa webpage (9.05 EUR/m$^2$ in 2016, 9.46 EUR/m$^2$ in 2017, 9.18 EUR/m$^2$ in 2018, 12.12 EUR/m$^2$ in 2019). Following the major increase observed in

2019 in relation to previous years this average then decreased to 11.63 EUR/m² in 2020. Once again, an upward trend line was observed with the gradual increase in price of rentals for long-term residence.

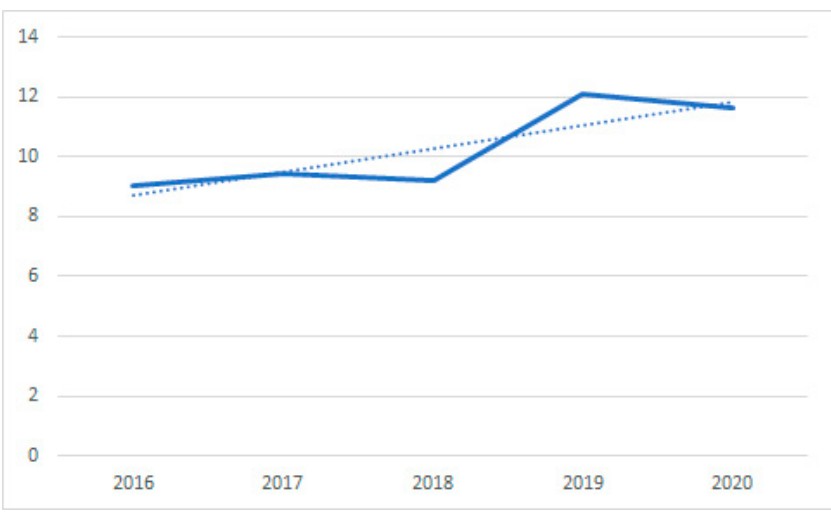

**Figure 4.** Value of land rent in the Casco Antiguo district. Source: the author, based on data obtained from [29].

*3.3. Socio-Cultural Analysis*

Finally, a socio-cultural analysis was carried out on the impact of the accommodation for temporary rental in the district, as this is one of the major effects of touristification increasing pressure on the resident population, and in turn on the environmental values considered in this section. Data were entered into the ArcGIS model in order to identify all the points representing an apartment featured on the Airbnb website for the city of Seville as of 02-10-2018 [27]. The total number of housing units per neighborhood was also obtained from Statistical Analysis [29] to contrast the percentage of accommodation used for tourist purposes.

Figure 5 shows the major concentration in the district of Casco Antiguo, with a high number of apartments devoted to tourist use. The same is observed in the Triana district, albeit on a smaller scale, as a very high concentration of tourist rentals is also found in the area by the river.

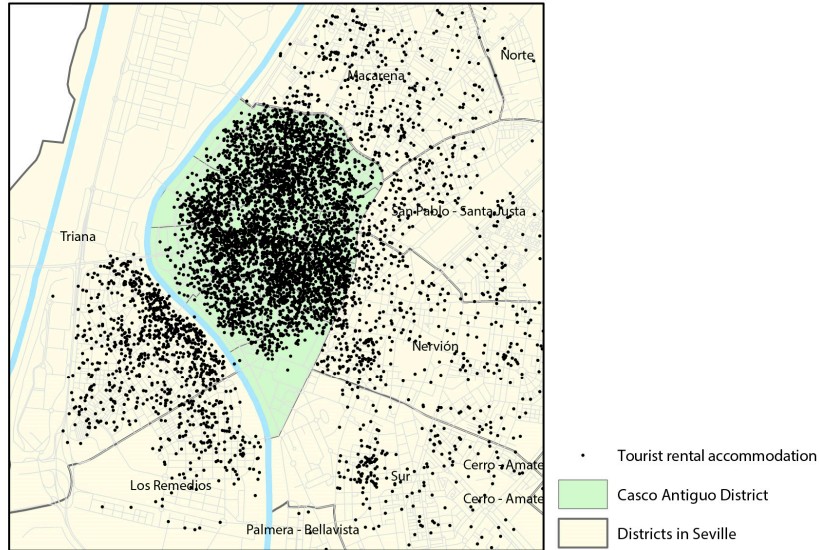

**Figure 5.** Tourist rentals in Seville. Source: the author, based on data obtained from [27].

In the specific case of the Casco Antiguo district the following analysis data were obtained: in 2018, 4840 of the 8815 tourist apartments recorded for the entire city of Seville were located in the district of Casco Antiguo, and this single district accounted for 50% of the short-term rental market. The total of housing units in the district, 27,187 in 2017 according to the Statistics Service, was also analyzed. Therefore, for the year 2018 the percentage of dwellings in the entire district dedicated to short-term rental represented 17.8% of the overall total for accommodation.

A detailed study of the individual neighborhoods of the district provided the following data:

Figure 6 shows the neighborhood with the highest percentage of tourist accommodation compared to permanent residence is that of Santa Cruz, where 37.29% and 427 of the neighborhood's 1145 dwellings were devoted to this temporary use. This was followed by the neighborhood of Alfalfa (32.35%) with a total of 2241 dwellings, 725 of which were tourist accommodation. In descending percentage order these neighborhoods are followed by the neighborhood of Arenal (24.89%), San Bartolomé (25.73%), Encarnación—Regina (21.76%), and a final group incorporating the rest of the neighborhoods, averaging approximately 15%.

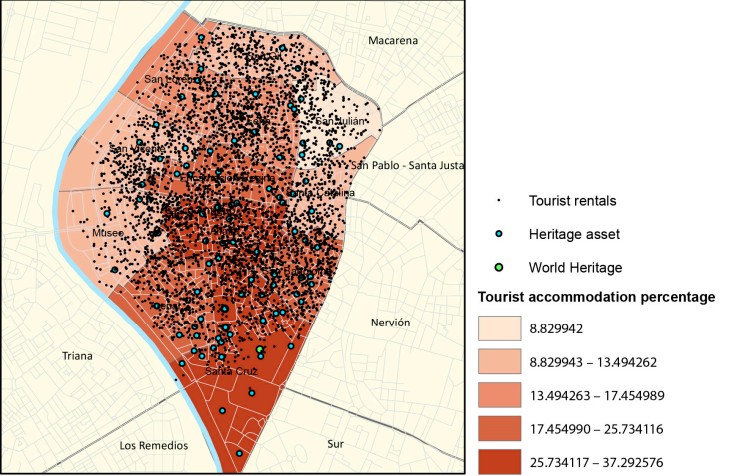

**Figure 6.** Percentage of tourist rentals in the district of Casco Antiguo compared to total number of dwellings. Source: the author, based on data obtained from [27,30,31].

In the Casco Antiguo district in Seville there is a total of 86 heritage assets according to the section on Spatial Reference Data, within the Instituto de Estadística y Cartografía de Andalucía [31]. The plan shows an equal distribution throughout the district, albeit with a greater heritage concentration in the center-south sector, coinciding with the neighborhoods able to accommodate a larger number of these tourist rentals.

Finally, as regards the defense of identity and appropriation, its evolution was studied by different bodies and collectives at the district level in order to present some sort of quantification of the importance of long-term residents defending a series of heritage values against great pressure from tourism. Between 2015 and 2017, a total of 11 registered bodies [30] was recorded. These included neighborhood associations promoting and calling for better living conditions, citizen participation, security, sustainable urban development, a solution to housing and location issues, and defending specific points. This was considered a sign of neighborhood activity within these bodies. One of these associations disappeared in 2018, bringing the total to 10, which remained the same until 2020, when it increased to 12. Therefore, there was a slight increase in the number of these collectives within the relative stability displayed in the period under study, reflecting an emerging concern for the current situation.

## 4. Conclusions

The analysis carried out in this article has presented a brief overview of the evolution of a series of reference aspects. These served to ascertain the presence or absence of gentrification processes, and more specifically, of touristification as the origin and root cause of the current situation. Although this analysis was carried out in three distinct blocks, the data obtained had to be cross-referenced as they interacted within a complex system where certain parts affected others, either directly or indirectly.

Firstly, a negative demographic effect was observed, decreasing at a local scale as well as within the specific district analyzed. When contextualized against other data, increases were also observed in both the rent prices for habitual residences and the land sale price. This suggests that the sector has undergone a process of revalorization, as when compared to other levels in the rest of the city, inflation was observed in the price per square meter due to its strategic location within the city center. An upward trend was also observed in the per-capita income for the different neighborhoods in the city, although this remained out of step at times with the development of the land market. This ultimately led to depopulation, with habitual residents being forced to move to areas in the city periphery, which offered more affordable housing rental and purchase prices.

At the same time, the use given to these dwellings has also transformed slightly, as many of these former homes had become an asset for financial gain within what is currently a macrosystem seeking to privatize the hospitality offer. This change in use is currently very aggressive as it takes place within an urban system that was previously consolidated to combine traditional activities and measured tourism but currently causes a situation of imbalance.

The appearance of neighborhood associations calling for the safeguarding of the identity and appropriation of these neighborhoods also reflects a collective feeling for the protection of a series of values which, as mentioned in the introduction, are eroded or in the worst case eventually disappear. The traditional population, deeply rooted with a historic memory, is being forced out to other sectors without being replaced by any other population. This is how we can see that there is no general process of gentrification at present in the historic center of Seville, but rather a specific touristification process. There is no process of land revalorization or improvement in the quality of life and purchasing power. Instead, we are observing a process of inflation which is leading to the depopulation of this sector of the city, taking with it the traditional identity and values to be replaced by a constantly moving rootless population, merely passing through and causing excessive exploitation of the city. Citizen participation has proven to be pivotal in addressing these issues, as the social aspect that is integral to the value of heritage sustainability resides within the very population inhabiting it, being perceived and valued by them [32]. The collaboration of societal representations in decision making is deemed necessary, as well as in the development and regulation of these processes, given their direct impact on the population as a whole.

Although tourism is one of the main economic pillars of Seville at present, this is a double-edged weapon as, if gone unchecked, it attacks the environmental values of a given urban enclave. This results in an excessive demand for cultural leisure and ultimately triggers a series of processes where negative connotations are attached to the different aspects reviewed. While gentrification, understood as a theoretical model, need not be a strictly negative process in society as it improves the conditions of a sector and renovates it, we find ourselves facing a very different scenario in this case. This is why the difference was established between both concepts from the beginning, principally based on their connotations and whether or not population is replaced through this process.

Thanks to the methodology presented in this research, the need for the production of a systematization becomes evident, with a general nature but also incorporating a specific component tailored to each population to thoroughly analyze these processes. This is crucial as these processes can be treated as constantly changing organisms that differ from

one another. Despite addressing the key aspects defining the heritage system today, this should be a future research avenue to be pursued in the short term.

These complex processes, sometimes unfolding over extended periods, require the establishment of conceptual methodologies that allow for the determination of their main parameters so that they can be studied and recognized [33]. The methodology presented in this research underscores the need for the development of a systematization, of a general nature, but also with a specific component for each population to thoroughly analyze these processes. This is because they can be treated as constantly evolving entities that differ from one another. Although the key aspects defining the heritage system today have been addressed, this should be a future research avenue to be tackled in the short term.

From the literature review and the emerging interest in gentrification in recent years, there arises a concern to develop sustainable or "green" gentrification models [13]. As this is an internationally relevant issue, predominantly affecting cities with a significant tourist influx, and where one of their economic pillars relies on this now excessive tourism, it becomes necessary for local administrations to establish sustainable models to control processes that end up harming the intrinsic heritage values of these cities [8,11]. In addition, there is a growing concern about the significant increase in tourist rental properties. While many municipalities have already begun to regulate them through the development of their own local policies, there is a need for further review and limitation [22].

Considering the evolution of the city in recent years, it is clear to see that it is in the process of recession, mainly because it has concentrated on economic development in a single aspect, tourism. The political actions of management entities should begin to ensure the safeguarding of these recessionary processes in future years [34]. The organisms in charge of Seville, such as the Ayuntamiento and its Gerencia de Urbanismo, should begin to take a stance and direct their policies against this situation, renovating and developing sustainable tourism models different from those seen since the late 20th century. Thus, a solution can be found for the present problems which affect the historic center of Seville.

**Author Contributions:** Conceptualization, G.H.-D., J.-M.A.-P. and J.R.-P.; methodology, G.H.-D.; software, G.H.-D.; validation, G.H.-D.; formal analysis, G.H.-D.; investigation, G.H.-D.; resources, G.H.-D.; data curation, G.H.-D.; writing—original draft preparation, G.H.-D.; writing—review and editing, G.H.-D.; visualization, G.H.-D.; supervision, G.H.-D., J.-M.A.-P. and J.R.-P.; funding acquisition, G.H.-D. and J.-M.A.-P. All authors have read and agreed to the published version of the manuscript.

**Funding:** The English revision of this text was funded through the Internationalization Grant by the University Institute of Architecture and Building Sciences in the framework of the Seventh Research and Transfer Plan of the University of Seville.

**Institutional Review Board Statement:** Not applicable.

**Informed Consent Statement:** Not applicable.

**Data Availability Statement:** Data are unavailable due to privacy.

**Conflicts of Interest:** The authors declare no conflict of interest.

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
