# Peer review of "Analysis of Touristification Processes in Historic Town Centers: The City of Seville"

_2673-8945, doi:10.3390/architecture4010003_

Round 1

Reviewer 1 Report

Comments and Suggestions for Authors

This paper, titled “Analysis of touristification processes in historic town centres: 2 application to the city of Seville”, presents an investigation regarding the effects of gentrification and touristification on the city centre of Seville.

The paper uses different methods and data to develop a series of analyses on the social, cultural and economic development of a historic area of Seville from 2015 to 2020.

In the introduction, there should be a more comprehensive reference to previous and present theoretical background and empirical research on the topic.

The methodology seems accurate. However, it lacks an overall acknowledgement and understanding of the research limits. It would be essential to include a further chapter/section with an in-depth discussion of the results, also in relation to adequate references and other studies.

The text follows a clear structure, but the conclusions are not effective. It would be important to rewrite them, further explaining the relevance of this study, not just about Sevilla, but to the research field.

Finally, the reference list is not acceptable. It is too short, focuses on two/three authors only, and is almost all Spanish. Several research projects have been conducted on the effects of gentrification and touristification on city centres in recent years, and several books and articles have been published internationally. The reference list needs to reflect a GLOBAL understanding of the state of the art of the research field.

I hope my comments are helpful to the author(s).

Comments on the Quality of English Language

The use of standard English should be improved. The terminology adopted often needs to be more specific and accurate.

Author Response

Firstly, thank you for your review of the article. Regarding your comments, the following fundamental aspects have been addressed:

  1. The discussion of results has been further explored, conducting a review of its international scope, as reflected in the introduction of the article through additional literature. Emphasis has also been placed on citizen participation and the development of methodologies and sustainable mechanisms to address this global issue.
  2. The bibliography has been reviewed, substantially expanded, and seamlessly incorporated into the article's narrative.

Thank you once again for your kind feedback.

Reviewer 2 Report

Comments and Suggestions for Authors

The article is focussed on Seville, yet a short report/cross reference to other spanish/European examples (maybe in the introduction and, perhapes, being recalled in the conclusions) would certainly give more soundness to the contribution, connecting it to a wider scope.

-The bibliography should be improved as well, following the previous point.

- maybe the conclusions could be improved by developing a little more the point "sustainable tourism models", possibly giving some examples.

Author Response

Firstly, thank you for your review of the article. Regarding your comments, the following fundamental aspects have been addressed:

  1. The bibliography has been reviewed, substantially expanded, and seamlessly incorporated into the article's narrative.
  2. The discussion of results has been further explored, conducting a review of its international scope, as reflected in the introduction of the article through additional literature. Emphasis has also been placed on citizen participation and the development of methodologies and sustainable mechanisms to address this global issue.

Thank you once again for your kind feedback.

Reviewer 3 Report

Comments and Suggestions for Authors

It was with a great interest that I read the article “Analysis of touristification processes in historic town centres: application to the city of Seville”. The subject of the study is very pertinent and up-to-date in a time of increasing attention paid to touristification and gentrification of the urban historic centres and the problems associated.

The manuscript complies with almost all the established norms for the articles of this publication. However, please note the followings:

In abstract the general purpose of the study is present but the concrete objectives should be clearly written, as well as the methodology and the main results.

- There is not a theoretical background to support the study. Some literature review is done in the introduction but the relevance of the topic deserves its own Literature Review section separate from the Introduction.

- The graphics must be reviewed. For example, Figure 1, it´s not necessary that in the horizontal axis appear Migratory pressure in all the observed years. Just put the years, Migratory pressure is the title. In Figure 2 a line chart is not the most indicated to this kind of data, a bar chart would be more appropriate.

Authors should review the way to reference because is not in accordance with the journal rules.

- It ss missing a discussion section of the results that establishes a bridge between what was found in this investigation and the literature.

- In conclusion section: what are the practical applications of this study? Limitations and future research should come here.

- The final reference list must be reviewed because do not follow the rules of the journal.

- The reference list must be improved.

Author Response

Firstly, thank you for your review of the article. Regarding your comments, the following fundamental aspects have been addressed:

  1. The potential introduction of a specific section for literature review has been taken into consideration. However, given the article's concise nature and the substantial modification it would entail to its primary objective, it has been ultimately decided to include the literature review within the Introduction section, as initially proposed.
  2. Figures 1 and 2 have been reviewed and adjusted in accordance with the suggested comments.
  3. The citation format of the article has been reviewed and adjusted to align with the guidelines of the journal.
  4. The discussion of results has been further explored, conducting a review of its international scope, as reflected in the introduction of the article through additional literature. Emphasis has also been placed on citizen participation and the development of methodologies and sustainable mechanisms to address this global issue.
  5. The bibliography has been reviewed, substantially expanded, and seamlessly incorporated into the article's narrative.

Thank you once again for your kind feedback.

Reviewer 4 Report

Comments and Suggestions for Authors

This study explores the impacts at several levels caused by touristification in historic town centres.  It uses multiple statistical data to question and reveal the negative impact of touristification on the sustainable development of  historic town centres. It provides meaningful case experience and critical counter-evidence to the myth that tourism drives urban development. Here are some suggestions:

 1. The literature review and theoretical discussion on the phenomenon of tourismization and gentrification in the redevelopment of historic town centres are insufficient. Therefore, this article lacks clear and in-depth theoretical questions. Finally, it only reveals the negative impact of tourismization in historic town centres without further profound theoretical dialogue.

2. Sustainability evaluation is the core aspect of this study for tourismization in historic town centres. It may be appropriate to make more supplements in the thesis title, literature review and research findings.

3. The conclusion can provide more complete answers and discussions to the aforementioned theoretical questions, and then put forward more specific policy recommendations.

Author Response

(The authors gave the same response as above.)

Round 2

Reviewer 1 Report

Comments and Suggestions for Authors

I want to thank the authors for amending the paper. Although some changes have been made, more is needed to improve the overall quality of the manuscript substantially.

The conclusion still needs to be clarified, and discussion on findings must be improved.

Also, the literature review still requires further investigation, and the reference list needs to be carefully developed.

Comments on the Quality of English Language

I want to thank the authors for amending the paper. Although some changes have been made, more is needed to improve the overall quality of the manuscript substantially.

The conclusion still needs to be clarified, and discussion on findings must be improved.

Also, the literature review still requires further investigation, and the reference list needs to be carefully developed.

Author Response

Firstly, thank you again for your review of the article. Regarding your comments, the following fundamental aspects have been addressed:

  1. The discussion of the article has been reviewed by the authors, concurring that its significance lies in the unique contribution of a methodology enabling the identification of these processes, as elaborated within the article itself.
  2. The bibliography has been reviewed, substantially expanded, and seamlessly incorporated into the article's narrative.

Thank you once again for your kind feedback.

Reviewer 3 Report

Comments and Suggestions for Authors

I would like to thank the authors for submitting the revised version of the manuscript. Some improvements have been made, but in my opinion they are not substantial.

- I still think that the literature review is insufficient, and I could even accept its inclusion in the introduction, but it needs to be solidified.

- Graph 1 has been altered, but the rest of the graphs have not. Too much information still appears on the horizontal axis, where only the years should be shown.

- The discussion of the results must be improved.

Author Response

Firstly, thank you again for your review of the article. Regarding your comments, the following fundamental aspects have been addressed:

  1. The bibliography has been reviewed, substantially expanded, and seamlessly incorporated into the article's narrative.
  2. All figures have been modified in accordance with your feedback.
  3. The discussion of the article has been reviewed by the authors, concurring that its significance lies in the unique contribution of a methodology enabling the identification of these processes, as elaborated within the article itself.

Thank you once again for your kind feedback.

Round 3

Reviewer 3 Report

Comments and Suggestions for Authors

I would like to thank the authors the corrections made to the manuscript, but there are still weaknesses that really need to be addressed.

From my point of view, the literature review is still insufficient given the subject matter. The reference list must be improved.

Author Response

Thank you for your review of the article. Regarding your comment, the following fundamental aspects have been addressed:

  1. The bibliography has been reviewed again, substantially expanded, and seamlessly incorporated into the article's narrative. 

Thank you once again for your kind feedback.